# Speech-enriched Memory for Inference-time Adaptation of ASR Models to Word Dictionaries

**Ashish Mittal[1,2], Sunita Sarawagi[2], Preethi Jyothi[2], George Saon[1], Gakuto Kurata[1]**
IBM Research[1]
IIT Bombay[2]
arakeshk@in.ibm.com

## Abstract

Despite the impressive performance of ASR models on mainstream benchmarks, their performance on rare words is unsatisfactory. In enterprise settings, often a focused list of entities (such as locations, names, etc) are available which can be used to adapt the model to the terminology of specific domains. In this paper, we present a novel inference algorithm that improves the prediction of state-of-the-art ASR models using nearest-neighbor-based matching on an inference-time word list. We consider both the Transducer architecture that is useful in the streaming setting, and state-of-the-art encoder-decoder models such as Whisper.

In our approach, a list of rare entities is indexed in a memory by synthesizing speech for each entry, and then storing the internal acoustic and language model states obtained from the best possible alignment on the ASR model. The memory is organized as a trie which we harness to perform a stateful lookup during inference. A key property of our extension is that we prevent spurious matches by restricting to only word-level matches. In our experiments on publicly available datasets and private benchmarks, we show that our method is effective in significantly improving rare word recognition.

## 1 Introduction

Rapid advancements in Automatic Speech Recognition (ASR) technologies have broadened their application in real-world settings. However, state-of-the-art ASR systems often fail to recognize named entities or critical rare words. For example, a chatbot we deployed in the medical domain backed by the state-of-art Whisper ASR model, consistently misrecognized *In patient* to *impatient*. Such situations call for *contextual biasing* whereby the transcription from ASR model is biased by a user-provided list of proper nouns (e.g., names from contact lists) or technical terms (e.g., drug names from medical dictionaries). The accurate transcription of such tokens could have a disproportionately

high impact on the ASR output's value. E.g., a spoken language understanding (SLU) module that uses ASR outputs might carry out the desired action as long as named entities in the spoken utterance are correctly transcribed.

Existing contextual biasing techniques typically demand additional parameterization, either via attention layers jointly trained with the ASR model (Le et al., 2021a) or via dictionary modules trained with the ASR model frozen (Sun et al., 2021). Inference-time adaptation to dictionaries has primarily been via the shallow fusion of an external LM (Kannan et al., 2018). However, such a late fusion overly biases the model to the external LM and incurs high decoding latency especially for streaming ASR. In our work, we address this gap and propose a new lightweight inference-time technique PRISM (short for "Ada**P**tation of AS**R** at **I**nference-time using **S**peech-enriched **M**emory") to adapt to predefined dictionaries at test-time with no additional training. PRISM is employed during beam-search decoding and used to adapt two state-of-the-art ASR architectures, encoder-decoder models (Chan et al., 2016) and Transducers (Graves, 2012) models.

First, we use a text-to-speech (TTS) system to synthesize speech for each dictionary entry. To the best of our knowledge, we are the first to use synthetic speech for contextual biasing and find that it facilitates improved matches compared to the use of text alone. Utilizing both speech and text in each dictionary entry, we create a memory mapping speech and text representations (keys) to textual tokens (values). For encoder-decoder models, the decoder states are a natural choice for the memory keys. However, for Transducer models, implementing memory keys requires more deliberation and careful handling of the alignment of acoustic and textual context (elaborated in §3.1).

During inference, we perform a $k$-nearest-neighbor (KNN) search over the memory to retrieve

the $k$ most similar dictionary entries. We define a KNN probability distribution over the next token that assigns higher probabilities to nearby memory entries and finally bias the model predictions via a linear interpolation with the kNN probability distribution. The memory-based matching is implemented using an efficient trie-based search. Notably different from prior work (Le et al., 2021a), we avoid triggering spurious matches in the memory after only a prefix match, by maintaining a backoff beam score that keeps track of the score assigned by the model to a suffix if it were not found in the trie. We show that such accounting is essential to avoid worsening baseline performance in the presence of a dictionary.

PRISM can be used with any existing pretrained encoder-decoder or Transducer ASR model without any retraining and incurs close to no delays during inference. Using PRISM, we obtain significant performance improvements with both Transducer and encoder-decoder ASR over diverse dictionaries, with up to $36\%$ relative reductions in entity-specific error rates on the well-known Librispeech ASR benchmark.

## 2 Related Work and Background

Prior work on contextual biasing of ASRs can be organized into three categories:

**Dictionary as external language model.** In the first category, the dictionary is viewed as an external LM. This is integrated with the ASR's output either via shallow fusion where ASR scores are combined with scores from an external LM during beam-search (Ravi et al., 2020; Liu et al., 2020; Le et al., 2021a; Gourav et al., 2021) or deep fusion where the model has access to an external LM during training (Le et al., 2021b,a). While shallow fusion is a popular scheme that allows for contextual information to be combined with any trained end-to-end ASR, it is also very sensitive to the weight assigned to the external LM and often helps ASR of contextual entities at the expense of overall ASR performance (Mittal et al., 2023).

**Training-time adaptation.** The second category consists of approaches that modify the ASR model during training to incorporate contextual information, often relying on attention-based mechanisms (Jain et al., 2020; Chang et al., 2021; Huber et al., 2021; Sathyendra et al., 2022; Sun et al., 2023a; Munkhdalai et al., 2023; Chan et al., 2023). Such a direct integration of contextual information

is usually more accurate than shallow fusion, but it comes with the added overhead of retraining the ASR model for every new dictionary to be integrated.

**Inference-time adaptation using dictionary.** The last category of approaches make use of a dictionary during decoding (Williams et al., 2018; Huang et al., 2020; Sun et al., 2021). PRISM falls in the last category with two key differences from prior work. We also synthesize speech, using TTS, for every dictionary entry, unlike prior work that only uses text from biasing lists/dictionaries. This facilitates improved matches to the dictionary terms at test time. We enforce word-level matches using a trie and additional beam state to avoid spurious prefix matches that could otherwise seriously distort baseline performance.

### 2.1 Background: ASR Models

Two prominent architectures for ASR are Transducers (Graves, 2012; Graves et al., 2013) popular when low latency and streaming (He et al.) are important, and encoder-decoder used in state-of-the-art ASR models such as Whisper (Radford et al., 2022). We present a brief overview of each.

**Transducers.** Transducers have three modules: (1) An audio encoder $\mathcal{M}_S$, (2) A text encoder $\mathcal{M}_L$, and (3) A joint network $\mathcal{M}_J$. Audio encoder $\mathcal{M}_S$ converts input speech $\boldsymbol{x}_1, \ldots, \boldsymbol{x}_T$ to the latent representations $\boldsymbol{h}_1, \ldots, \boldsymbol{h}_T$ using an RNN (Cho et al., 2014) or Transformers (Vaswani et al., 2017) or Conformer (Gulati et al., 2020). The text encoder $\mathcal{M}_L$ converts each text prefix $\boldsymbol{y}_u = y_1, \ldots, y_{u-1}$ to the contextual latent representation $\boldsymbol{g}_u$. The joint network $\mathcal{M}_J$ combines the latent representations $h_t$ and $g_u$, followed by a softmax layer that outputs the distribution $P_\theta(y|\boldsymbol{g}_u, \boldsymbol{h}_t)$ over vocabulary $\mathcal{V}$ plus the blank symbol $\varnothing$. Since no explicit alignments between $\boldsymbol{x}_i$ and $\boldsymbol{y}_u$ is provided, log-likelihood of the target sequence $\boldsymbol{y}$ is computed during training by marginalizing over all possible alignments $(t, u)$ using an efficient forward-backward algorithm (Graves et al., 2013). During inference, beam-search is used to find the best possible $(t, u)$ alignment (Graves, 2012; Saon et al., 2020).

**Encoder-decoder.** These models encode the audio in the same way as above but the decoder converts the text prefix $\boldsymbol{y}_u$ and audio using cross-attention on the audio states at each decoding step. Thus, the alignment of text and audio is implicit in the cross

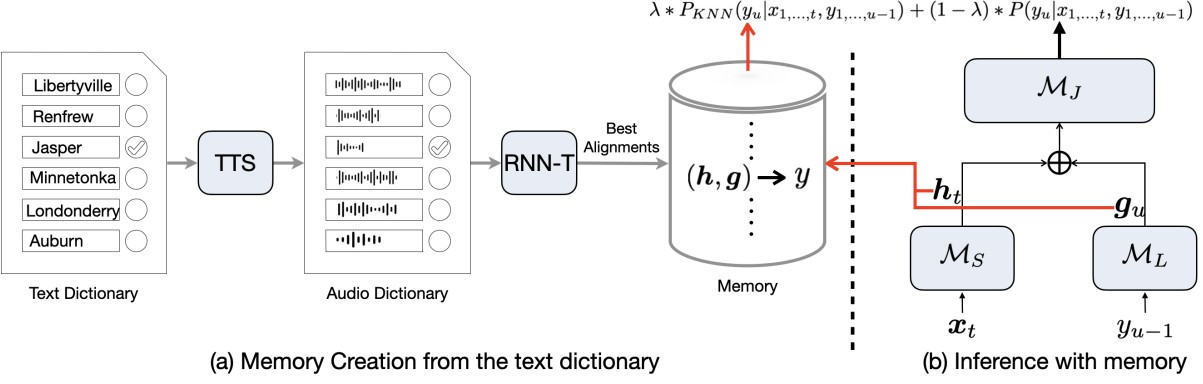

$$\lambda * P_{KNN}(y_u|x_{1,...,t}, y_{1,...,u-1}) + (1 - \lambda) * P(y_u|x_{1,...,t}, y_{1,...,u-1})$$

(a) Memory Creation from the text dictionary

(b) Inference with memory

Figure 1: Overview of our approach for a Transducer ASR model.

attention. At decoding step $u$, a cross attention from decoder to the encoder states $h_1, \ldots, h_T$ provides a fixed length vector $\mathbf{e}_u$ which is fused with self-attention layers on the decoder to get an updated decoder state $\mathbf{d}_u$. The decoder state followed by a softmax layer yields $P_\theta(y|\mathbf{d}_u, h)$. Inference happens via beam search.

## 3 Contextual biasing in PRISM

We are given a trained ASR model $\mathcal{M}$ and a list of isolated word phrases $\tilde{D} = \{\tilde{\boldsymbol{y}}^1, \ldots, \tilde{\boldsymbol{y}}^k\}$ at inference time. PRISM uses $\tilde{D}$ to first create a memory structure $N$. Subsequently, for each subsequent input speech $\boldsymbol{x}$, we perform a memory augmented inference to bias its text to the words in $\tilde{D}$. As stated earlier, the challenge is that recognition of words not in $\tilde{D}$ should not worsen in the process. In order to not adversely impact the streaming and latency performance of $\mathcal{M}$, PRISM integrates with existing ASR decoding algorithms at the token-level, by just modifying the token distribution at each step of decoding.

An efficient mechanism proposed in the NLP literature for sequence to sequence models is k-nearest neighbor (KNN) language models (Khandelwal et al., 2019; Guu et al., 2020; Yogatama et al., 2021). A key insight of KNN LMs is that similar decoder contexts lead to similar next token distribution. This motivates the creation of memory in the form of a list of key-value pairs with context as key and next-token as value.

However, applying this principle to ASR models raises several challenges. First, we are given as input only a list of isolated word phrases, and context is missing both in terms of the missing audio input, and an overall sentence context in which the phrase lies. We partly offset for the lack of audio input

by resorting to TTS to generate synthetic audio for each phrase. A second challenge is that for Transducer models, the design and extraction of context to form the memory keys is significantly harder, since they do not follow an encoder-decoder architecture. Finally, a flat memory in the unit of tokens causes several spurious matches with pieces of a whole phrase during decoding. For example, a dictionary entry for "Intuit" could spurious matches to the "In" or "it" tokens of this word. We therefore restrict to allow only whole phrase matches. To implement such whole phrase matches over multiple steps in existing token-level decoding algorithms, we organize the memory as a trie and maintain state in the beam on the prefix of the trie matched so far. Additionally, we maintain a backoff beam score that keeps track of the model score for retraction unless the match completes a full word. We first present how we create the memory and then discuss our method of interpolating default model token distribution $P_\theta$ with lookups from the memory.

### 3.1 Memory Creation

Since the provided list $\tilde{D}$ is in text, we use a TTS engine to create audio for the corresponding phrases as described in Figure 1(a). With this, we obtain a target domain dataset $\tilde{\mathcal{D}}_T$: $\{\ldots, (\tilde{\boldsymbol{x}}^i, \tilde{\boldsymbol{y}}^i), \ldots\}$ which is devoid of any context in which these utterances are being used. We convert $\tilde{\mathcal{D}}_T$ into a trie-based memory of key-value pairs at the token-level as follows.

**Memory keys for Transducers** For Transducers, the internal representations to index are audio encoder $\mathcal{M}_S$ state $h_t$ and the text encoder $\mathcal{M}_L$ state $g_u$ for each non-blank prediction $y_u$ as they describe both the acoustic state and contextual language model state for the predicted token. For

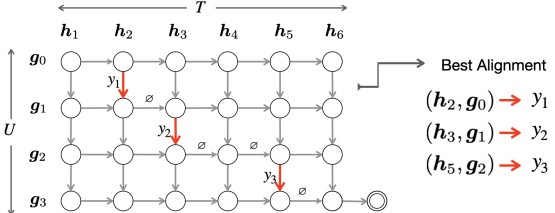

Figure 2: For transducer models the best alignments from the decoding lattice are used to generate the memory.

each utterance $\tilde{x}^i$ in $\tilde{\mathcal{D}}_T$, we trace the lattice of the highest probability beam to find the best valid alignment $A^i$ that consists of a sequence of $(\boldsymbol{h}_t, \boldsymbol{g}_u, y_u)$ corresponding to non-blank predictions as described in Figure 2. We augment the beam search algorithm to constrain the output vocabulary to the target sequence $\tilde{\boldsymbol{y}}^i$ and $\varnothing$. For instance, if the target sequence is "Minnetonka", and the predicted sequence in the beam is "Minne", then for the subsequent expansion of the beam, the output vocabulary is constrained to either 't' or $\varnothing$. To further nudge the model to predict the character 't', we incrementally add extra probability to the constrained token over $\varnothing$. We find that this algorithm provides better alignments as opposed to finding the best path in the output lattice using a Viterbi algorithm.

**Memory keys for Encoder-decoder Models** We use teacher forcing to constrain the predicted output to the ground truth. For each inference step the memory is the output $\mathbf{d}$ after the last cross-attention block in the decoder. The value for the memory consists of the next token as per the ground truth.

## 3.2 Memory Augmented Decoding

We present the framework for memory augmented inference that applies both for Transducers and encoder-decoder models. Both models use some form of beam search during inference, and we modify each decoding step of beam-search as follows:

At each decoding step $i$, we have a current hypothesis $\boldsymbol{y}_u = y_1, \ldots, y_u$, a decoder state $\mathbf{d}$, a beam score $s$, and a next token distribution $P_\theta(y|\boldsymbol{y}_u, \boldsymbol{h})$ from the model. From the memory, that is organized as a trie, we maintain state in the form of the trie node $N$ that has matched a suffix of $\boldsymbol{y}_u$, call it $\boldsymbol{z}_u = y_{u-\ell} \ldots y_u$ where $\ell$ is the depth of $N$. The suffix could be empty, in which case $N$ is root of the trie. We define a query $q_i$ from the decoder state $\mathbf{d}$, and use that to find the K-

nearest neighbors (K is a hyperparameter) from the children of $N$ as a set $\mathrm{NN}(N, q_i) = \{(k_r, v_r)\}$ denoting key and token id of the matched neighbors. The distance function between $q_i$ and $k_r$ is model dependent. We use thresholds on the distances and it is possible for the neighbor set to be empty.

When the neighbor set $\mathrm{NN}(N, q_i)$ is non-empty we define a distribution over the next token from the memory lookup as:

$$P_{\mathrm{knn}}(y|q_i, N) \propto \sum_{(k_r, y) \in \mathrm{NN}(N, q_i)} \exp(-\mathrm{dist}(q_i, k_r))$$
(1)

The above distribution is interpolated with the token distribution from the model as

$$P(y|\cdot) = \begin{cases} (1 - \lambda_i)P_\theta(y|\cdot) + \lambda_i P_{\mathrm{knn}}(y|\cdot) & \text{if } y \in N \\ P_\theta(y|\cdot) & \text{otherwise} \end{cases}$$
(2)

The KNN coefficient $\lambda_i$ can be considered as a hyper-parameter, but we find that deriving the value of the KNN Coefficient dynamically using a simple linear function on the minimum distance yields improved performance.

$$\lambda_i = 1 - \min_{(k_r, v_r) \in \mathrm{NN}(N, q_i)} (\mathrm{dist}(q_i, k_r)) \quad (3)$$

This function has the effect of increasing KNN weight when there is a good match in the dictionary.

Unlike in previous work, one crucial step we take to avoid spurious matches with the memory, is to only allow the memory scores to accumulate for full word matches. Without it, we observe that when the actual word deviates from the dictionary word after a prefix match, the hypothesis still carries the extra probability and can lead to spurious recognition (e.g, twin becomes twen, if the dictionary has the word twente as shown in Figure 3). In order to avoid such spurious matches and enforce word-level match only, we maintain on the beam in addition to the beam score $s$, a default score $s_\theta$ that tracks the model score of the suffix if the current suffix $\boldsymbol{z}_u$ of $\boldsymbol{y}_u$ were not to match the trie. If the next token of $\boldsymbol{y}$ does not match a child of $N$, we reset the beam score $s$ to $s_\theta$. If we reach the end of the trie at a decoding step, the complete word has matched and we update $s_\theta$ to be the beam score $s$. We will show in the ablation that such a step was essential to avoid penalizing recognition accuracy of words that do not match any full word in the dictionary. The overall algorithm appears in Algorithm 1 assuming a generic beam-search algorithm for encoder-decoder model. For the Transducer

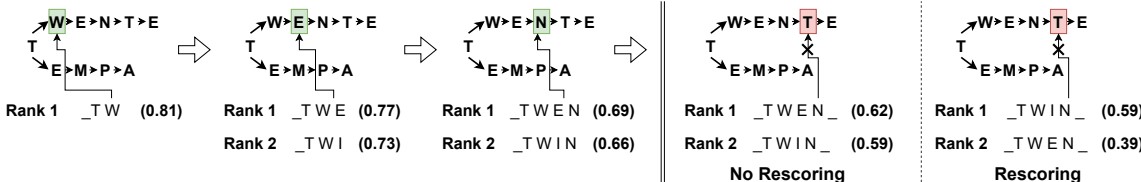

Figure 3: Step-by-step decoding using PRISM for the reference word **twin**. For this example, the numbers on the right of each step are the probability of that hypothesis. For each correct match, the probability is assumed to be 0.9. For the match in step 2, the probability before and after the interpolation are 0.95 and 0.6 respectively. With rescoring, the interpolation probabilities are replaced by the model probability, which downgrades the hypothesis.

model, while the overall framework is the same, the details are slightly different. We provide a detailed algorithm for Transducers in Appendix B.

---

**Algorithm 1** PRISM Inference

---

1: **Input**: Encoder state $h$, beam width $W$, max. decode length $I$, Trie memory root $N$
2: **Output**: Most likely hypothesis in beam
3: Beam = $\{(y = \phi, \mathbf{d} = \text{initial state}, N, s = 1, s_\theta = 1)\}$

4: **for** $i = 1 \dots I$ **do**
5:     $A = \{\}$
6:     **for** $(y, \mathbf{d}, N, s, s_\theta) \in$ Beam **do**
7:         $P_\theta(y|\cdot), \mathbf{d} = \mathcal{M}(h, y, \mathbf{d})$
8:         knn, $\lambda$ = Near nbrs($\mathbf{d}, N$, thresholds)
9:         $P_{\text{knn}}$ = KNN-probs(knn, $\mathbf{d}$) (Eqn: 1)
10:         **for** $k \in \mathcal{V}$ **do**
11:             $s_\theta(y + k) = s_\theta(y)P_\theta(k)$
12:             $s(y + k) = s(y)(\lambda P_\theta(k) + (1 - \lambda)P_{\text{knn}}(k))$
13:             **if** $k \notin N.$next **then**
14:                 $s(y + k) = s_\theta(y + k)$ (forget knn scores if incomplete trie match)
15:             **if** $k \in N.$end **then**
16:                 $s_\theta(y + k) = s(y + k)$ (reset)
17:             $A = A \cup \{(y + k, \mathbf{d}, N.\text{next}(k), s(y + k), s_\theta(y + k))\}$
18:     Beam=PruneAndRecombineHyps$(A, W)$
19: **return** Sorted(Beam) =0

---

## 4 Experiments

We present an extensive comparison of PRISM with existing methods on both state-of-the-art Transducer (Graves et al., 2013) and encoder-decoder (Chan et al., 2016) ASR models spanning three different dataset settings.

**Models and Implementation Details.** For the encoder-decoder models, we use two publicly-available Whisper models (Radford et al., 2022): Whisper-base(74M parameters) and Whisper-small (244M parameters). Whisper was trained on over 600K hours of weakly labeled training data mined from the web. We do not do any finetuning of the Whisper model, and only alter the beam-search

decoder to incorporate PRISM. We used a beam-width of 10 in all the Whisper experiments. The Transducer model (56M parameters) is trained on a proprietary dataset of 56K hours of labeled speech. This model takes 240-dimensional log-Mel filter-bank audio features as its input and predicts probability distributions over an output vocabulary of 45 English characters and a blank character. We elaborate in Appendix A.

**Memory.** We use the FAISS library (Johnson et al., 2019) to index and efficiently search through embeddings. Recall that we index embeddings from both the audio and text encoder for the Transducer model. We use a state-of-the-art VITS (Kim et al., 2021) system from Coqui TTS (coqui.ai, 2023) to synthesize audio for dictionary entries. We set the thresholds for the audio match to 0.4 and text match to 0.05. Note that the same threshold is used across all datasets spanning 9 different dictionary lists. The text-based threshold is strict, thereby only allowing for matches that the language model readily permits. However, the audio-based threshold is more relaxed allowing for more variability in the audio matches to accommodate varying pronunciations, noise in the audio, etc. With the Whisper models, we use an unnormalized embedding-based matching with a distance threshold of 250 and 1000 for the base and small models respectively. (Normalized distances did not perform as well.) We further apply a min-max normalization to these unnormalized distances to use as a dynamic KNN coefficient for the interpolation (outlined in Section 3.2). For all the experiments, we use the four nearest neighbors while searching the index.

**Evaluation and Comparisons.** Apart from computing an overall word error rate (WER), we also compute an entity WER (E-WER) over the spans covered by dictionary entities in the test utterances.

We compare against three existing techniques.

| | test-clean | test-other |
|---|---|---|
| TCPGen (Sun et al., 2023b) | | |
| Whisper Baseline | 5.2 / 16.3 | 10.5 / 31.4 |
| Whisper + TCPGen | 4.8 / 12.5 | 10.1 / 26.5 |
| Relative Reduction | 7.7 / **23.3** | 3.8 / 15.6 |
| PRISM (Ours) | | |
| Whisper Baseline | 4.9 / 16.1 | 10.2 / 30.9 |
| Whisper+PRISM | 4.4 / 13.0 | 9.6 / 25.7 |
| Relative Reduction | **10.2** / 19.3 | **5.9 / 16.8** |

Table 1: Comparison of WER/E-WER on Whisper with TCPGen on base.en model. Whisper + TCPGen numbers are reproduced from (Sun et al., 2023b), along with their Whisper baseline numbers. PRISM outperforms TCPGen on both the test sets from a superior baseline.

| | test-clean | test-other |
|---|---|---|
| TCP Gen (Sun et al., 2021) | | |
| RNN-T | 5.5 / 18.7 | 15.3 / 42.9 |
| RNN-T+TCPGen | 4.9 / 13.9 | 14.0 / 35.0 |
| Relative Reduction | 10.9 / 25.7 | 8.5 / 18.4 |
| Deep Contextual RNN-T (Le et al., 2021a) | | |
| RNN-T | 3.7 / 14.1 | 9.6 / 30.6 |
| RNN-T+Deep Cont. | 3.0 / 8.5 | 8.5 / 20.5 |
| Relative Reduction | **17.8 / 39.7** | **11.5** / 33.0 |
| PRISM (Ours) | | |
| RNN-T | 3.5 / 13.3 | 9.9 / 32.4 |
| RNN-T+PRISM | 3.1 / 9.3 | 9.0 / 20.6 |
| Relative Reduction | 11.4 / 30.1 | 9.1 / **36.0** |

Table 2: Comparison of WER/E-WER with TCPGen and Deep Contextual RNN-T for RNN-T models. PRISM obtains a 30% reduction in E-WER without needing an external language model used in Deep Contextual RNN-T and outperforms TCPGen on all metrics.

1. TCPGen (Sun et al., 2021) learns a distribution over the output vocabulary that is limited by a prefix tree constructed from the dictionaries. TCPGen has a parameterized module, jointly trained with the frozen ASR model, to estimate weights signifying how much contextual bias is required to decode the current token. TCPGen has been used both with the Transducer model and Whisper (Sun et al., 2021, 2023b).
2. Deep Contextual RNN-T (Le et al., 2021a) consists of learnable trie-based biasing modules and is specific to the Transducer model.

Unlike PRISM, we note that both TCPGen and Deep Contextual RNN-T techniques require a joint training of biasing modules with the ASR models.

**Overview of Experiments.** We show ASR adaptation experiments across multiple scenarios involving three types of datasets:

1. **Librispeech (publicly-available) Dataset:** We evaluate on the popular Librispeech ASR benchmark (Panayotov et al., 2015) using dictionaries from prior work (Sun et al., 2021; Le et al., 2021a). These experiments are detailed in §4.1.
2. **Enterprise Dataset:** We show experiments on a real consumer-facing chatbot application in the medical domain (in §4.2). We evaluate on real speech from users covering five different intents and dictionaries of varying sizes.
3. **Entity-rich Dataset:** Due to the absence of publicly-available entity-rich datasets, we create a new benchmark with sentences that are dense in entities such as locations, person names and medicine names (detailed in §4.3). This is a synthetic dataset created using TTS that

lends itself nicely to detailed ablations by enabling fine-grained control on dictionary size, types of entities, etc. We will publicly release this benchmark to facilitate further research into dictionary-based ASR adaptation.

### 4.1 Evaluation on Librispeech

We evaluate on the standard test-clean and test-other splits of Librispeech (Panayotov et al., 2015). For dictionaries, we borrow the experimental protocol used for TCPGen Sun et al. (2021). Biasing lists comprise 200K rare words from the Librispeech training set. Dynamic dictionaries are constructed for each test utterance. Rare words in the test transcript are augmented with randomly sampled rare words to create 1000-word dictionaries per test utterance. We reuse the same dictionaries used by Sun et al. (2021) for a fair comparison. Additionally for PRISM, we create supporting audio for each dictionary entry using TTS.

Table 1 shows how PRISM improves over Whisper and how it compares to TCPGen. Whisper+TCPGen WERs are reproduced as it appears in Sun et al. (2023b), along with their reported Whisper baseline numbers. We observe that PRISM yields superior WER and E-WER reductions on both test-other and test-clean. We achieve significant reductions compared to TCPGen, despite starting with a much stronger Whisper baseline; in fact, our baseline numbers are almost as good as the reported Whisper+TCPGen numbers.

| | Category - 1 | Category - 2 | Category - 3 | Category - 4 | Category - 5 | Average |
|---|---|---|---|---|---|---|
| RNN-T Baseline | 32.9 / 26.3 | 38.8 / 32.6 | 48.6 / 26.6 | 125.0 / 83.3 | 33.0 / 40.0 | 55.7 / 41.8 |
| Whisper (small) | 32.9 / 21.2 | 77.6 / 28.6 | 45.2 / 24.5 | 116.7 / 58.3 | 53.2 / 44.1 | 65.2 / 35.3 |
| Peng et al. (2023) | 19.2 / 13.1 | 63.3 / 26.5 | 57.1 / 40.0 | 58.3 / 33.3 | 58.7 / 30.9 | 51.3 / 28.8 |
| PRISM | **26.0 / 15.8** | **26.5 / 26.5** | **40.0 / 20.0** | **16.7 / 8.3** | **28.2 / 29.4** | **27.5 / 20.0** |

Table 3: Comparison of WER/E-WER on the internal medical domain dataset using an RNN-T and Whisper(small) baseline. PRISM outperforms all the baselines on most categories for this noisy short-form internal dataset halving the WER and E-WER.

Table 2 presents results on the Transducer ASR comparing PRISM with both TCPGen and Deep Contextual RNN-T baselines. PRISM outperforms both baselines on test-other in terms of relative WER and E-WER reductions and performs comparably to Deep Contextual RNN-T (and better than TCPGen) on test-clean. For both Whisper and Transducer models, it is noteworthy that PRISM achieves comparable WER reductions with minimal overhead, compared to TCPGen and Deep Contextual RNN-T that both require retraining.

## 4.2 Evaluation on Enterprise Dataset

We evaluate PRISM using data from a deployed chatbot in the medical domain, interacting with real users. We test on utterances drawn from five intents/categories with keywords specific to each category. Category-1 contains utterances with 6 domain-specific keywords (e.g., *representative*, *authorization*), Category-2 has 4 keywords (e.g., *pharmacy*), Category-3 has 4 keywords (e.g., *acupuncture*), Category-4 has 2 keywords (*inpatient*, *outpatient*) and Category-5 has 30 keywords (e.g., *mammogram*, *chemotherapy*).

Table 3 shows evaluation results for all five categories, using both Transducer and Whisper baseline models, along with PRISM applied to the Transducer model. For a test utterance in Category-4 with the keywords *inpatient* and *outpatient*, the baseline Transducer model predicts *impatient* which is a more common word in the training set, while Whisper predicts *in patience*. PRISM fixes such errors and dramatically reduces the baseline Transducer WER from 83.3 to 8.3 over 20 test utterances. Other categories also significantly benefit from using PRISM. The Whisper baseline is prone to hallucinate with irrelevant content, especially when the input audio is short; Appendix C shows some examples.

Additionally, in Table 3, we compare PRISM with a very recently proposed technique from Peng et al. (2023) that is applicable only to the Whisper model. Whisper supports the use of prompts with the decoder state to bias the predictions towards the dictionary entries during decoding.[1] Note that this method is applicable in settings where the number of contextual entities is small, as the decoder state is prompted on these entities. Hence, we compare this technique with PRISM for the enterprise dataset that contains a small number of keywords in each category. We observe that PRISM significantly outperforms the technique from Peng et al. (2023) on our enterprise benchmark. The average WER/E-WER from Peng et al. (2023) is 51.3/28.8 compared to 27.5/20.0 obtained using PRISM.

## 4.3 Evaluation on Entity-rich Dataset

We create a new entity-rich, synthetic benchmark spanning different categories of commonly occurring entities viz. locations, person names and medical drugs. We use VITS TTS from the Coqui toolkit (coqui.ai, 2023) to create both entities and sentences for this benchmark. We use different speakers for the dictionary entities and the test utterances to mimic real settings where these speakers are disjoint. This benchmark is released[2] to encourage fair comparisons of dictionary-based adaptation techniques. Prior work on contextual biasing either uses proprietary datasets or public lists (like from Librispeech), but the latter is neither domain-specific nor dense in entities. Table 4 provides a brief description of our dataset, along with entity types and sample sentences containing these entities. Table 5 shows results from both Transducer and Whisper models, with and without PRISM. We note the high baseline E-WER, especially for Drugs and Location (big), indicating that the entities are difficult for the baseline models to

---

[1]More information is available at `https://github.com/openai/whisper/discussions/117#discussioncomment-3727051`.

[2]https://github.com/AshishMittal/PRISM

| Entity | #Entities | #Sentences | Sample Entities | Sample Sentences |
|---|---|---|---|---|
| Location (small) | 1000 | 567 | Chicopee, Erlanger Minnetonka | I live in Chicopee. Erlanger and Minnetonka are gorgeous. |
| Location (big) | 2818 | 1628 | Albemarle, Saratoga Merthyr Tydfil | The route passes through Albemarle and Saratoga. Merthyr Tydfil is gorgeous. |
| Person Names | 4348 | 4220 | Beaufort, Zebadiah Eberhard | Zebadiah always makes sure to include others. Eberhard loves to skydive. |
| Drugs | 2874 | 2874 | Aclovate, Primidone Fluothane | Is Primidone safe to use during pregnancy? What is the dosage for Fluothane? |

Table 4: Entity-rich dataset.

| | | Location (small) | Location (big) | Person Names | Drugs | Average |
|---|---|---|---|---|---|---|
| RNN-T | Baseline | 39.1 / 70.0 | 37.6 / 67.7 | 24.7 / 19.0 | 36.8 / 93.5 | 34.6 / 62.3 |
| | PRISM | **27.1 / 41.0** | **29.0 / 45.1** | **24.3 / 17.6** | **33.5 / 77.0** | **28.5 / 45.1** |
| Whisper (small) | Baseline | 24.4 / 45.1 | 23.5 / 45.1 | 12.8 / 15.3 | 23.3 / 82.2 | 21.0 / 46.9 |
| | PRISM | **18.8 / 31.8** | **18.2 / 32.2** | **12.1 / 14.1** | **17.6 / 53.9** | **16.7 / 33.0** |

Table 5: Comparison of WER/E-WER on different entities from Entity-rich dataset. PRISM consistently outperforms the baseline RNN-T and Whisper models yielding upto 20% and 30% reductions in WER and E-WER respectively.

| | Location (Big) | Drugs |
|---|---|---|
| PRISM | 29.0 / 45.1 | 33.5 / 77.0 |
| - Dynamic Coefficient | 36.4 / 63.5 | 35.3 / 85.4 |
| - Trie | 35.4 / 53.9 | 36.5 / 80.3 |
| - Rescore at false Trie exit | 32.0 / 52.9 | 36.8 / 79.3 |
| Audio-only Index | 53.3 / 43.9 | 39.2 / 73.3 |
| Text-only Index | 115.2 / 99.8 | 139.1 / 96.8 |

Table 6: WER/E-WER comparison after removing core components from PRISM for RNN-T. Using audio-only index for RNN-T yields better E-WER, but the overall WER degrades because of the unconstrained hallucinations due to lack of language model context.

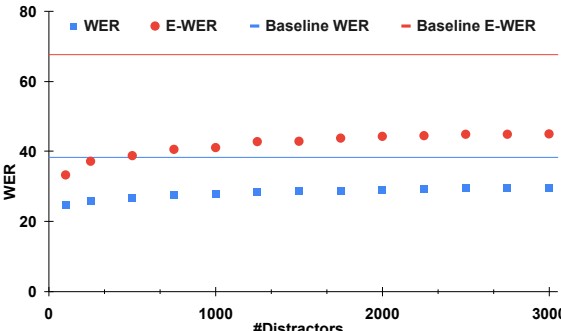

Figure 4: Comparison of WER vs. #distractors in the dictionary for Location (Big).

accurately transcribe. PRISM yields significant WER/E-WER reductions for all entity types.

## 4.4 Ablations and Analysis

We use the synthetic benchmark to present an ablation of various design choices that are critical to the success of PRISM. Table 6 shows how PRISM fares on Location (Big) and Drug entity types if: 1) we omit the dynamic KNN coefficient and use a static coefficient instead, 2) we use a flat memory rather than a trie-based memory and, 3) we remove the rescore feature in the trie (described in Section 3.2) that encourages full-word matches. We observe that each of these design choices were important. Note that disabling the rescore feature

in the trie significantly degrades performance. This feature is absent from all prior trie-based biasing techniques (Sun et al., 2021; Le et al., 2021a; Sun et al., 2023b). Table 6 also shows the importance of indexing both audio and text embeddings. Overall WERs degrade with using an audio-only index (due to the lack of language modeling context), while both overall WERs and E-WERs drastically worsen with using a text-only index.

Figure 4 shows how PRISM behaves with varying dictionary sizes for Location (Big). For each test utterance, we identify the gold entities from the reference text and add a fixed number of randomly-sampled distractors to reach the desired dictionary

| Dataset | No of Entities | Average Tokens | Memory |
|---|---|---|---|
| Location (small) | 1000 | 10.8 | 50 MB |
| Location (large) | 2818 | 10.7 | 138 MB |
| Person Names | 4348 | 7.0 | 126 MB |
| Drugs | 2874 | 11.3 | 132 MB |

Table 7: Memory requirement of various dictionaries.

size. We see that PRISM is steady in its performance and does not degrade in the presence of large number of distractors.

### 4.5 Memory and Compute Overhead of PRISM

We compute the memory occupied by our dictionary for various entities in the entity-rich dataset in Table 7. It is worth noting that the sizes of the dictionaries are influenced by the number of tokens (characters or BPE units) present in the words or phrases within the dictionary entries. While the Drugs entity dictionary contains fewer entries, the average token count per entry tends to be larger compared to the Person Names and Location entity dictionaries.

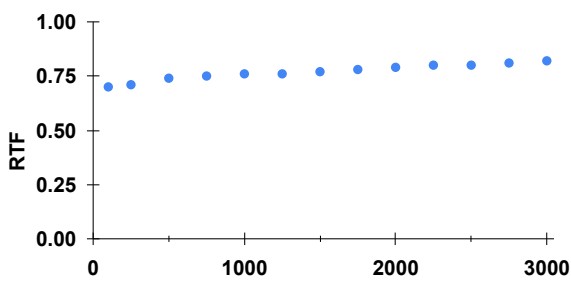

Figure 5: Comparison of RTF vs. #distractors in the dictionary. Baseline RTF is 0.5

Next, we study the overhead of using PRISM during inference. Analytically, the complexity of the beam search algorithm for RNN-T models is $O(cBUT)$, where $c$ is the cost of the forward pass, $B$ is the beam size, $T$ is the number of audio frames in the input and $U$ is the maximum length of the output. PRISM method adds a lookup cost to memory in each step of the beam expansion. If the cost of lookup is $p$, then the overall complexity would become $O((c + p)BUT)$. We use the state-of-the-art nearest neighbor index from Johnson et al. (2019) which can do a fast look-up to minimize the latency.

To highlight the minimal computational overhead incurred by PRISM, we compute RTF (real-time factor) scores i.e., the time it takes to decode 1 second of audio, using our Transducer model with

dictionaries of varying sizes for Location (Big). Figure 5 shows that PRISM scales well and incurs minor overheads with larger dictionaries.

## 5 Conclusion

We present PRISM, an efficient inference-time dictionary-based ASR adaptation technique that uses synthesized speech for dictionary entries to facilitate improved matches. PRISM includes an efficient trie-based memory lookup algorithm, with effective fallback scoring mechanisms to prevent spurious matches. PRISM can augment standard beam-search decoders in both Transducer and Whisper encoder-decoder ASR models. We demonstrate the effectiveness of PRISM with experiments across diverse dictionaries containing rare words and named entities. We obtain significant relative WER reductions of up to $36\%$ on entities, yielding an overall WER reduction close to $10\%$, on the challenging test-other split in Librispeech benchmark. We also achieve superior performance on evaluation sets drawn from an enterprise setting, with overall and entity error rates almost reduced by a factor of 2 (i.e., relative reductions of $40\%$ to $55\%$ over the baseline ASR systems).

## 6 Limitations

We highlight two limitations of our current work:

- PRISM relies on TTS systems to create audio representations for dictionary keywords. For many rare words, TTS might not provide the correct pronunciation which can lead to mismatches with the actual words during inference. In ASR settings involving accented speech and low-resource languages, this difference can be further exacerbated due to the lack of robust TTS engines.
- Memory and runtime requirements for PRISM are dependent on the size of the memory. This can be further optimized by quantizing the representations.

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

## A  Implementation details

The model consists of 6 bidirectional LSTM layers of hidden dimensionality 640 in the acoustic encoder and a single unidirectional 768-dimensional LSTM layer in the text encoder. Both acoustic and text encoders are further projected down to 256 dimensions using separate projection layers, and a joint network combines these 256-dimensional embeddings via a simple element-wise product, followed by a tanh non-linearity. A final softmax projection layer is used to produce probability distributions over an output vocabulary consisting of 45 non-blank English characters and a blank character. This model uses log-Mel filterbank audio features augmented with delta and double-delta spectral features (Mason and Zhang, 1991; Picone, 1993), resulting in 240-dimensional input features spanning 20ms audio segments. As is standard practice, audio augmentations such as speed and tempo perturbations (Park et al., 2019) are applied as well. The Transducer models are trained for 20 epochs using AdamW (Loshchilov and Hutter, 2017) with a maximum learning rate of 5e-4.

## B  Detailed PRISM algorithm for RNN-T

Here we describe PRISM algorithm for the Transducer model that follows the same structure as Algorithm 1 defined earlier. We extend alignment length synchronous decoding algorithm (Saon et al., 2020) for transducers with PRISM. The key addition to the generic PRISM for the transducer models are the steps in lines 13-15. Lines 13-15 show the addition of the $\varnothing$ hypothesis to move the audio state ($h_t$) in the RNN-T lattice. The rest of the hypothesis expansion with non-blank characters remains the same as the generic PRISM algorithm. Note that PRISM can be adapted to other search algorithms such as the time-synchronous algorithm described in (Graves, 2012) using similar steps.

For the transducer models PRISM uses separate indexes for audio and text embeddings. In the implementation, both indexes are queried separately with their respective thresholds. The common values obtained from the indexes are combined to form a single output over which KNN probabilities (described in eqn 1) are computed.

## C  Examples

In this section, we show anecdotal examples of how PRISM improves over the baseline models. Table

---

**Algorithm 2** Pseudo-code for Memory Augmented ALSD for RNN-T

1: **Input**: Encoder state $h$, beam width $W$, max. decode length $I$, Trie memory root $N$
2: **Output**: Most likely hypothesis in beam
3: Beam $= \{(\boldsymbol{y} = \phi, g_0 = \text{initial state}, N, s = 1, s_\theta = 1)\}$
4: $F = \{\}$
5: **for** $i = 1...T + I$ **do**
6:    $A = \{\}$
7:    **for** $(\boldsymbol{y}, g_{u-1}, N, s, s_\theta) \in$ Beam **do**
8:       $u = |\boldsymbol{y}|$
9:       $t = i - u + 1$
10:      **if** $t > T$ **then**
11:         continue
12:      $P_\theta(y|\cdot), g_u = \mathcal{M}(\boldsymbol{h}_t, \boldsymbol{y}, g_{u-1})$
13:      $s(\boldsymbol{y}) = s(\boldsymbol{y})P_\theta(\phi)$
14:      $s_\theta(\boldsymbol{y}) = s_\theta(\boldsymbol{y})P_\theta(\phi)$
15:      $A = A \cup \{(\boldsymbol{y}, g_{u-1}, N, s(\boldsymbol{y}), s_\theta(\boldsymbol{y}))\}$
16:      **if** t$==T$ **then**
17:         $F = F \cup \{(\boldsymbol{y}, s(\boldsymbol{y}))\}$
18:      knn, $\lambda$ = Near nbrs($g_u, h_t, N$, thresholds)
19:      $P_\text{knn} = $ KNN-probs(knn, $g_u, h_t$) (Eqn: 1)
20:      **for** $k \in \mathcal{V}$ **do**
21:        $s_\theta(\boldsymbol{y} + k) = s_\theta(\boldsymbol{y})P_\theta(k)$
22:        $s(\boldsymbol{y} + k) = s(\boldsymbol{y})(\lambda P_\theta(k) + (1 - \lambda)P_\text{knn}(k))$
23:        **if** $k \notin N.\text{next}$ **then**
24:          $s(\boldsymbol{y} + k) = s_\theta(\boldsymbol{y} + k)$ (forget knn scores if incomplete trie match)
25:        **if** $k \in N.\text{end}$ **then**
26:          $s_\theta(\boldsymbol{y} + k) = s(\boldsymbol{y} + k)$ (reset)
27:        $A = A \cup \{(\boldsymbol{y} + k, g_u, N.\text{next}(k), s(\boldsymbol{y} + k), s_\theta(\boldsymbol{y} + k))\}$
28:    $B = $ PruneAndRecombineHyps($A, beam\_size$)
29: **return** sorted($F$) =0

9 shows the examples from Whisper (base.en) baseline and PRISM adapted Whisper for librispeech dataset. Note that with PRISM many challenging dictionary words get corrected due to robust KNN-based matching. Next, in Table 10, we show anecdotal examples from the Location dataset. Many challenging acoustic words such as *Kissimmee, Burnhamonsea*, etc. are recognized correctly by PRISM. We also show some examples from the challenging Drugs dataset in Table 11, where hard-to-spell medical entities are recognized correctly.

In Table 12 and 13, we show the anecdotal examples from the Location and Drugs benchmarks for transducer models, showing great effectiveness in improving the E-WER.

| | |
|---|---|
| Reference | Representative. |
| Whisper (small) | For every single day. |
| Reference | Pharmacy. |
| Whisper (small) | Guys have a great winter every one. |
| Reference | Representative. |
| Whisper (small) | I will finish up before we open all right Thank you O. K. just so they can remove of. Thank you very much I remember before they married they were in the disable. family came home seen at the camp alarmed guruji we do not want to ruin the family in sight. |

Table 8: Hallucination examples on the Internal Dataset.

| | |
|---|---|
| Reference | the europe they had come from lay out there beyond the irish sea europe of strange **tongues** and **valleyed** and **woodbegirt** and **citadelled** and of **entrenched** and **marshalled races** |
| Baseline | the europe they had come from lay out there beyond the irish sea europe of strange **tongues** and valid and woodbeagert and citadeld and of **entrenched** and **marshalled races** |
| PRISM | the europe they had come from lay out there beyond the irish sea europe of strange **tongues** and valid and **woodbegirt** and **citadelled** and of **entrenched** and **marshalled races** |
| Reference | american school boys read with emotions of horror of the **albigenses** driven beaten and killed with a **papal legate** directing the **butchery** and of the **vaudois** hunted and **hounded** like beasts as the effect of royal **decree** and they yet shall read in the history of their own country of scenes as terrible as these in the **exhibition** of injustice and **inhuman** hate |
| Baseline | american schoolboys read with emotions of horror of the albiginses driven beaten and killed with a **papal legate** directing the **butchery** and of the vaudoua hunted and **hounded** like beasts as the effect of royal **decree** and they yet shall read in the history of their own country of scenes as terrible as these in the **exhibition** of injustice and **inhuman** hate |
| PRISM | american schoolboys read with emotions of horror of the **albigenses** driven beaten and killed with a **papal legate** directing the **butchery** and of the **vaudois** hunted and **hounded** like beasts as the effect of royal **decree** and they yet shall read in the history of their own country of scenes as terrible as these in the **exhibition** of injustice and **inhuman** hate |
| Reference | **fauchelevent grumbled** more to himself than to jean valjean |
| Baseline | fortunately von **grumbled** more to himself than to jean valjean |
| PRISM | **fauchelevent grumbled** more to himself than to jean valjean |
| Reference | and i **sicut terra sine aqua** |
| Baseline | and i sycote terrasin **aqua** |
| PRISM | and i **sicut terra** sin **aqua** |

Table 9: Examples on Librispeech with Whisper (base.en) and PRISM. The text in the boldface are the entities present in the index.

| Reference | I live in **Burnhamonsea** |
| --- | --- |
| Baseline | I live in Bonamancee |
| PRISM | I live in **Burnhamonsea** |
| Reference | **Irvine**, **Salt Lake City**, and **Sandown** are important cities |
| Baseline | I have unsolved **Lake City** and sundown my important cities |
| PRISM | **Irvine**, **Salt Lake City**, and **Sandown** are important cities |
| Reference | Telephone was invented in **Kissimmee** |
| Baseline | Telephone was invented in kiss me |
| PRISM | Telephone was invented in **Kissimmee** |
| Reference | Distance between **Massapequa Park** and **Beltsville** is small |
| Baseline | The distance between Massacre **Park** and belzelis mall |
| PRISM | The distance between **Massapequa Park** and **Beltsville** is small |
| Reference | The distance between **Failsworth** and **Atascocita** is quite large |
| Baseline | The distance between Faresworth and Ataskasesia is quite large |
| PRISM | The distance between **Failsworth** and **Atascocita** is quite large |
| Reference | My house is near **Aiken** and **Bloxwich** |
| Baseline | My house is no Akon and Blockswich |
| PRISM | My house is no **Aiken** and **Bloxwich** |

Table 10: Examples on Location (entity-rich dataset) with Whisper (small) and PRISM. The text in the boldface are the entities present in the index.

| Reference | What are the common brand names of **Sudafed** |
| --- | --- |
| Baseline | What are the common brand names of Sudest |
| PRISM | What are the common brand names of **Sudafed** |
| Reference | What is the recommended dosage for **Flavoxate** |
| Baseline | What is the recommended dosage for Flavix 8 |
| PRISM | What is the recommended dosage for **Flavoxate** |
| Reference | What should I do if I experience side effects from **Epoprostenol Sodium** |
| Baseline | What should I do if I experience side effects from Epipersonal **Sodium** |
| PRISM | What should I do if I experience side effects from **Epoprostenol Sodium** |
| Reference | Can **Diltiazem** be used in combination with other medications |
| Baseline | Candleshays can be used in combination with other medications |
| PRISM | Can **Diltiazem** be used in combination with other medications |

Table 11: Examples on Drugs (entity-rich dataset) with Whisper (small) and PRISM. The text in the boldface are the entities present in the index.

| | |
|---|---|
| Reference | **Preston** is near to the beach but far from **Warrensburg** |
| Baseline | **Preston** is near to the beach but far from war and spoke |
| PRISM | **Preston** is near to the beach but far from **Warrensburg** |
| Reference | People from **Murfreesboro** are generous |
| Baseline | People from murphy sproy are generous |
| PRISM | People from **Murfreesboro** are generous |
| Reference | **Cypress** and **Alliance** are very clean |
| Baseline | Cyprus and **Alliance** are very clean |
| PRISM | **Cypress** and **Alliance** are very clean |
| Reference | I live between **Canterbury** and **Wichita** |
| Baseline | I live between Canterbay and which attack |
| PRISM | I live between **Canterbury** and **Wichita** |
| Reference | The branch of the bank is located in **Los Gatos** |
| Baseline | The branch of the bank is located in **Los** Angeles |
| PRISM | The branch of the bank is located in **Los Gatos** |

Table 12: Examples on Location (entity-rich dataset) with RNN-T Baseline and PRISM. The text in the boldface are the entities present in the index.

| | |
|---|---|
| Reference | What are the possible withdrawal symptoms if **Fabrazyme** is abruptly discontinued |
| Baseline | What are the possible withdrawal symptoms if february time is abruptly discontinued |
| PRISM | What are the possible withdrawal symptoms if **Fabrazyme** is abruptly discontinued |
| Reference | Is there a generic version of **Dexmedetomidine Hydrochloride** |
| Baseline | Is there a generic version of dexperate attempting **hydrochloride** |
| PRISM | Is there a generic version of **Dexmedetomidine Hydrochloride** |
| Reference | What is the mechanism of action of **Colazal** |
| Baseline | What is the mechanism of action of colousal |
| PRISM | What is the mechanism of action of **Colazal** |
| Reference | Are there any lifestyle modifications or precautions to consider while taking **Raloxifene** |
| Baseline | Are there any lifestyle modifications or precautions to consider while taking right so I think |
| PRISM | Are there any lifestyle modifications or precautions to consider while taking **Raloxifene** |

Table 13: Examples on Drugs (entity-rich dataset) with RNN-T Baseline and PRISM. The text in the boldface are the entities present in the index.