# OpenReview forum: "Speech-enriched Memory for Inference-time Adaptation of ASR Models to Word Dictionaries"
_EMNLP/2023/Conference — EMNLP 2023 Main_

### Official Review · Reviewer_emcA · 2023-07-24

**Soundness:** 4

**Excitement:**

4: Strong: This paper deepens the understanding of some phenomenon or lowers the barriers to an existing research direction.

**Paper Topic And Main Contributions:**

Topic: contextual biasing for ASR systems.

The authors proposed an inference time adaptation algorithm inspired by KNN LM to bias ASR model decoding towards a predefined rare word dictionary, while do not worsen the performance on words not in that rare word dictionary by using a trie for exact word match and using a backup beam. The proposed method leverages speech synthesis systems, and is training-free, which are the main novelty compared to other contextual biasing approaches.

Experiments show that the proposed algorithm works on both encoder-decoder architecture (the Whisper model) and Neural Transducers (the RNN-T model)

In addition, the release of the entity-rich dataset could be a valuable contribution to the community as well.


**Questions For The Authors:**

1. Based on description in section 4.1 and table 1, I'm confused by line 455 "despite starting with a much stronger Whisper baseline", how can it be much stronger if both this paper and Sun et al 2023 are using base.en.
2. In Table 2, the RNN-T baseline numbers for the three approaches are different, e.g. on test-clean, WERs are 5.5 for Sun et al. 3.7 for Le et al. and 3.5 for this work. How can that be the case, as the RNN-T model is the same, and contextual biasing modules are not added for baseline.
3. For Whisper, there should be a very simple way to achieve contextual biasing. As indicated in figure 1 of the original whisper paper [1] and executed in the AVSR section of [2] (note that [2] is published after the submission of this paper), is to use a predefined dictionary in the prompt to the decoder can bias the model to output words from or related to the words in the predefined dictionary. Using prompt for contextual biasing seems particularly suitable for the experiment in section 4.2 where the evaluation is on Enterprise dataset, because the vocab. is small, and therefore the overhead to context length is small.

[1] Radford, Alec, et al. "Robust speech recognition via large-scale weak supervision." International Conference on Machine Learning. PMLR, 2023.

[2] Peng, Puyuan, et al. "Prompting the Hidden Talent of Web-Scale Speech Models for Zero-Shot Task Generalization." Interspeech, 2023.

**Reasons To Accept:**

1. The idea of organizing memory as a trie and use a backoff beam to avoid spurious matches is elegant, and is shown to avoid worsening the performance on words not in the rare word dictionary
2. the approaches works for both encoder-decoder models and Transducer models. While it's not as straightforward to apply their approach on Transducer models, the authors found tricks that make it work
3. the evaluation is comprehensive - on publicly available dataset, proprietary data set in medical domain, and synthetic entity-rich dataset

**Reasons To Reject:**

1. Latency of the proposed approach, and comparison with other approaches are not studied. I suspect that this approach is faster than neural nets based adaptation approaches that are compared in the paper, but this needs to be shown quantitatively.
2. whether the proposed approach is superior to Le at al 2021a is not clear to me, because PRISM and Le et al. 2021a are only compared once in table 2, and PRISM performs worse than Le et al.2021a. However, the paper claims that "PRISM achieves superior WER reductions" in line 467.

**Reproducibility:**

3: Could reproduce the results with some difficulty. The settings of parameters are underspecified or subjectively determined; the training/evaluation data are not widely available.

**Reviewer Confidence:**

4: Quite sure. I tried to check the important points carefully. It's unlikely, though conceivable, that I missed something that should affect my ratings.

**Typos Grammar Style And Presentation Improvements:**

Line 35: state-of-art -> state-of-the-art

Line 195: … perform a memory augmented inference to… -> … perform memory augmented inference to…

---

> ### Author Rebuttal · Authors · 2023-08-29
>
> We thank the reviewer for their insightful comments. We address the reviewer’s questions below.
>
> Q: `Latency compared with Neural methods.`
>
> A: We compare the latency of PRISM with our implementation of Contextual RNN-T paper by measuring the Real-Time Factor (RTF). RTF is the time it takes to decode 1 second of audio.
>
> | Dictionary Size | Contextual RNN-T | PRISM |
> |-----------------|------------------|-------|
> |             100 |             0.71 |  0.70 |
> |             250 |             0.76 |  0.71 |
> |             500 |             0.82 |  0.74 |
> |            1000 |             0.88 |  0.76 |
> |            2000 |             0.92 |  0.79 |
>
>
> We show that the latency of PRISM is comparable to the latency of neural methods such as Contextual RNN-T [1] (precursor to Deep Contextual RNN-T). Computing attention for each of the context entities becomes expensive as the dictionary size increases, similar to PRISM.
>
> [1] Jain, Mahaveer, et al. "Contextual RNN-T for open domain ASR." arXiv preprint arXiv:2006.03411 (2020).
>
> Q: `PRISM better than Le at al 2021a.`
>
> A: Since the model from Le at al 2021a is not publicly available, we could not evaluate it on the entity-rich datasets. We will fix our claim that PRISM is comparable to Le at al 2021a in terms of performance despite being an inference time adaptation to the dictionary and not requiring expensive model fine-tuning.
>
> Q: `Discrepancy on Whisper results on base.en from Sun et al and PRISM for Librispeech dataset.`
>
> A: While the evaluation on the Librispeech test set is performed using the same Whisper base.en model, some key hyperparameters such as beam size, patience, etc. are not described in Sun et al. For PRISM, we use a beam size of 10 and patience=1.0 for the evaluation. This would explain the discrepancy between results.
>
> Q: `Different WERs on Librispeech for RNN-T Models in Sun et al, Le et al 2021a, and PRISM?`
>
> A: While a uniform model for training RNN-T models is appealing, different papers have different model architectures for training the RNN-T models.
>
> Sun et al: Has a 4-layer BiLSTM encoder with 1024 hidden dimensions.
>
> Le et al 2023a: Has a 20-layer streamable low-latency Emformer model for the acoustic network. (83M parameters)
>
> PRISM: Has a 6-layer BiLSTM encoder with 6024 hidden dimensions. (56M parameters)
>
> Each paper has a different model architecture with a different number of parameters used for training the baseline model. This would explain the discrepancy in the baseline results using RNN-T for the Librispeech test sets.
>
> Q: `Achieve contextual biasing in Whisper using AVSR biasing technique shown in Peng et al 2023.`
>
> A: We thank the reviewer for this exciting experiment. We show a comparison of Peng et al. 2023 with PRISM on the Enterprise dataset.
>
> |                              | Category - 1 | Category - 2 | Category - 3 | Category - 4 | Category - 5 | Average     |
> |------------------------------|--------------|--------------|--------------|--------------|--------------|-------------|
> | RNN-T Baseline               |  32.9 / 26.3 |  38.8 / 32.6 |  48.6 / 26.6 | 125.0 / 83.3 |  33.0 / 40.0 | 55.7 / 41.8 |
> | Whisper (small)              |  32.9 / 21.2 |  77.6 / 28.6 |  45.2 / 24.5 | 116.7 / 58.3 |  53.2 / 44.1 | 65.2 / 35.3 |
> | Whisper (small) + Peng et al |  19.2 / 13.1 |  63.3 / 26.5 |   57.1/ 40.0 |  58.3 / 33.3 |  58.7 / 30.9 | 51.3 / 28.8 |
> | PRISM                        |  26.0 / 15.8 |  26.5 / 26.5 |  40.0 / 20.0 |   16.7 / 8.3 |  28.2 / 29.4 | 27.5 / 20.0 |
>
> We observe that PRISM significantly outperforms the Peng et al. method on our Enterprise benchmark. The average WER/E-WER from Peng et al. is 51.3/28.8 compared to 27.5/20.0 obtained using PRISM.

---

### Official Review · Reviewer_Kmjd · 2023-08-04

**Soundness:** 4

**Excitement:**

4: Strong: This paper deepens the understanding of some phenomenon or lowers the barriers to an existing research direction.

**Paper Topic And Main Contributions:**

The authors present an inference method for streaming and non-streaming ASR to decode rare words. The method is based on TTS and k-NN search to bias the decoding of rare entities towards a set of pre-defined words.


**Questions For The Authors:**

For Table3: please say if PRISM is used with RNN-T or Whisper and please explain why E-WER is sometimes lower than WER. E-WER is supposed to contain rare words that are harder to decode.

**Reasons To Accept:**

The methodology is solid: large set of experiments, multiple datasets, comparison to the state of the art. Even if the improvement compare to Deep Contextual, the method is novel and bold (using TTS and k-NN was a risky solution as dealing with speech embedding is always harder than using text embeddings).

**Reasons To Reject:**

The method is quite hard to follow. I am still not sure to really understand how everything works. Especially section 3.1 please be more specific on how do you get the alignments.

**Reproducibility:**

4: Could mostly reproduce the results, but there may be some variation because of sample variance or minor variations in their interpretation of the protocol or method.

**Reviewer Confidence:**

3: Pretty sure, but there's a chance I missed something. Although I have a good feel for this area in general, I did not carefully check the paper's details, e.g., the math, experimental design, or novelty.

---

> ### Author Rebuttal · Authors · 2023-08-29
>
> We thank the reviewer for their insightful comments. We address the reviewer’s questions below.
>
> Q: `Please be more specific on how do you get the alignments.`
>
> A: For encoder-decoder based Whisper model, the alignments are obtained by teacher forcing the decoder output during beam search. We store the representations of the decoder state as the key in the memory, while the value is the token that is to be predicted for a given dictionary item.
>
> For RNN-T, this is slightly more involved as a blank character is also part of the model’s output, which makes teacher forcing difficult. Our solution is to incrementally nudge the model’s probability towards the token in the sequence. For example, if the dictionary word is “As_ pi_ rin” and the current generation so far is “As_ pi_”, we increase the likelihood of token “rin” by a constant factor so that its prediction is favoured as opposed to other tokens. We find that this closely mimics the decoding process and the representations obtained are closely aligned with PRISM’s inference.
>
> We will make this clearer in the paper. Thanks for pointing this out.
>
> Q: `Why E-WER is sometimes lower than WER? E-WER is supposed to contain rare words that are harder to decode.`
>
> A: In the internal datasets, we often encounter audio files that are short utterances with long pauses and silences in the beginning. This leads to hallucinations in both the RNN-T and Whisper models. In particular, Whisper significantly hallucinates as demonstrated in the two examples below.
>
> *Example #1*
>
> Reference: representative
>
> Whisper output: I will finish up before we opena ll right. Thank you. O.K. Just so they can remove oh thank you very much. I remember before they married they were in the disabled family came home seen at the camp alarmed guruji we do not want to ruin the family in sight.
>
> *Example #2*
>
> Reference: 	pharmacy
>
> Whisper output: 	guys have a great winter everyone.
>
> This results in an inflation in overall WERs. Entity WERs, on the other hand, are computed only over spans containing the desired entities in the alignment. This is the main reason why E-WERs are sometimes lower than the overall WERs.

---

### Official Review · Reviewer_KQzm · 2023-08-11

**Soundness:** 4

**Excitement:**

4: Strong: This paper deepens the understanding of some phenomenon or lowers the barriers to an existing research direction.

**Missing References:**

https://arxiv.org/abs/2301.00066

**Paper Topic And Main Contributions:**

This work applied memory-augmented decoding (kNN retrieval) to ASR systems (both transducers and enc-dec). This method requires a speech/text representation to map to target token sequences; the authors propose to use TTS to obtain speech inputs to the model. Results show consistent improvements over three datasets (librispeech, medical domain dataset, and an entity-rich dataset).

**Questions For The Authors:**

Is it possible to do an ablation on TTS vs real audio for creating the memory?

Is this method limited to word-based biasing lists? Can it extend to phrases?

What is the added computational cost?

**Reasons To Accept:**

The proposed method is novel. Components are well-motivated and supported by ablations (i.e. dynamic knn weight, trie-based memory). Method is applied to both whisper encoder-decoder and rnn-t.

**Reasons To Reject:**

Comparisons to other contextual biasing methods is limited. Results from two other "inference-time adaptations using dictionary" methods are shown for comparison, but the authors do not apply TCPGen or deep contextual biasing rnn-t to their own models/experimental setup. Perhaps I have misunderstood something, but it appears that we cannot directly compare these methods based on tables 1 and 2.

**Reproducibility:**

3: Could reproduce the results with some difficulty. The settings of parameters are underspecified or subjectively determined; the training/evaluation data are not widely available.

**Reviewer Confidence:**

3: Pretty sure, but there's a chance I missed something. Although I have a good feel for this area in general, I did not carefully check the paper's details, e.g., the math, experimental design, or novelty.

---

> ### Author Rebuttal · Authors · 2023-08-29
>
> We thank the reviewer for their insightful comments. We address the reviewer’s questions below.
>
> Q: `Running TCPGen and Deep Contextual RNN-T on our benchmarks?`
>
> A: Since none of the above techniques are open-source, we weren’t able to run the models on TCPGen and Deep Contextual RNN-T. We tried the implementation of Contextual RNN-T (precursor to Deep Contextual RNN-T paper) and trained it on SWB 300H dataset, but we did not observe performance improvements with dictionary biasing. In the original paper, the model is trained on 8000 hours of proprietary data which indicates getting these models to work requires significant resources in terms of labelled data and computation.
>
> Q: `Is it possible to do an ablation on TTS vs real audio for creating the memory?`
>
> A: Collecting real audio for an entity-rich dataset was not feasible within the rebuttal timeline.  However, we perform a proxy experiment using the open-source Librispeech dataset. We use Librispeech’s training set to extract representations for dictionary items and evaluate using the same protocol described in the paper.  We show the WER/E-WER results on the test-clean split.
>
> |                       | WER / E-WER |
> |-----------------------|-------------|
> | Baseline              | 3.5 / 13.3  |
> | PRISM (w/ TTS Audio)  | 3.3 / 11.1  |
> | PRISM (w/ real audio) | 3.3 / 11.5  |
>
> For this experiment, the training set of Librispeech did not have all the dictionary entries present which reduces the size of the dictionary. For the TTS audio experiment, we only use those entries that are present in the training set to ensure a fair comparison. As shown in the results, PRISM is robust to representations from real audio as well. We will add these results to the paper.
>
> Q: `Is this method limited to word-based biasing lists? Can it extend to phrases?`
>
> A: The biasing list entries can be phrases in our implementation. In fact, 3 of the 4 dictionaries for the entity-rich dataset have phrases with multiple words.
>
> | Dataset | %Entities with  multiple words |
> |---|---|
> | Location (small) | 32.9 |
> | Location (large) | 31.5 |
> | Drugs | 15.9 |
>
> We plan to release the entity-rich dataset to promote further research in this area.
>
> Q: `What is the added computational cost?`
>
> A: We first compute the memory occupied by our dictionary for various entities in the entity-rich dataset.
> | Dataset          | #No of entities | #Average Tokens | Memory |
> |------------------|-----------------|-----------------|--------|
> | Location (Small) |            1000 |            10.8 |  50 MB |
> | Location (Large) |            3000 |            10.7 | 138 MB |
> | Person Names     |            4347 |             7.0 | 126 MB |
> | Drugs            |            2874 |            11.3 | 132 MB |
>
> It's worth noting that the sizes of the dictionaries are influenced by the number of tokens (characters or BPE units) present in the words or phrases within the dictionary entries. While the 'Drugs' entity dictionary contains fewer entries, the average token count per entry tends to be larger compared to the 'Person Names' and 'Location' entity dictionaries."
>
> However, as a fraction of the model size, the additional memory footprint due to the dictionary is modest because unused entries of the trie are not loaded in memory.  For example for the Location (Large) dictionary, we report the memory footprint of the baseline RNN-T model, and RNN-T with PRISM dictionary.
>
> Baseline RNN-T: 	810.5 MB
>
> RNN-T with PRISM: 	850.5 MB
>
> Analytically, the complexity of the beam search algorithm for RNN-T models is given as O(B * T * U * C), where B is the beam size, T is the number of audio frames in the input, U is the max length of the output, and C is the cost of the forward pass. PRISM method adds a lookup cost to memory in each step of the beam expansion. If the cost of lookup is P, then the overall complexity would become O(B * T * U * (C + P)).
> We use the state-of-the-art nearest neighbor index from Johnson et al [1], which can do a fast look-up to minimize the latency.
>
> [1] Johnson, Jeff, Matthijs Douze, and Hervé Jégou. "Billion-scale similarity search with GPUs." IEEE Transactions on Big Data 7.3 (2019): 535-547.
>
> Thanks for pointing out the missing reference, we will add the citation.

---

### Official Review · Reviewer_wrWb · 2023-08-12

**Soundness:** 3

**Excitement:**

4: Strong: This paper deepens the understanding of some phenomenon or lowers the barriers to an existing research direction.

**Paper Topic And Main Contributions:**

In this paper authors propose a new method named "AdaPtation of ASR at Inference-time using Speech-enriched Memory" (PRISM), which improves ASR results by adapting to predefined dictionaries during inference. PRISM leverages k-nearest neighbor (KNN) and text-to-speech (TTS), and experimental results show its effectiveness.

**Questions For The Authors:**

Question A: What are the model complexities for models in Table 1&2, in terms of computation and memory?

Question B: In Table 2, are the results of RNN-T+TCPGen and RNN-T+Deep Cont. taken from literature, or regenerated?


**Reasons To Accept:**

This proposed PRISM is applicable to different ASR model architectures, and it doesn't require to retrain ASR models. It also runs efficiently during inference. There are well-designed experimental validation and ablation studies presented.

**Reasons To Reject:**

The experimental section could be made stronger with more details added (see questions in the section below).

**Reproducibility:**

4: Could mostly reproduce the results, but there may be some variation because of sample variance or minor variations in their interpretation of the protocol or method.

**Reviewer Confidence:**

3: Pretty sure, but there's a chance I missed something. Although I have a good feel for this area in general, I did not carefully check the paper's details, e.g., the math, experimental design, or novelty.

---

> ### Author Rebuttal · Authors · 2023-08-29
>
> We thank the reviewer for their insightful comments. We address the reviewer’s questions below.
>
> Q: `What are the model complexities for models in Tables 1&2 and, in terms of computation and memory?`
>
> A: We first compute the memory occupied by our dictionary for various entities in the entity-rich dataset.
>
> | Dataset          | #No of entities | #Average Tokens | Memory |
> |------------------|-----------------|-----------------|--------|
> | Location (Small) |            1000 |            10.8 |  50 MB |
> | Location (Large) |            3000 |            10.7 | 138 MB |
> | Person Names     |            4347 |             7.0 | 126 MB |
> | Drugs            |            2874 |            11.3 | 132 MB |
>
> It's worth noting that the sizes of the dictionaries are influenced by the number of tokens (characters or BPE units) present in the words or phrases within the dictionary entries. While the 'Drugs' entity dictionary contains fewer entries, the average token count per entry tends to be larger compared to the 'Person Names' and 'Location' entity dictionaries."
>
> However, as a fraction of the model size, the additional memory footprint due to the dictionary is modest because unused entries of the trie are not loaded in memory.  For example for the Location (Large) dictionary, we report the memory footprint of the baseline RNN-T model, and RNN-T with PRISM dictionary.
>
> Baseline RNN-T: 	810.5 MB
>
> RNN-T with PRISM: 	850.5 MB
>
> Analytically, the complexity of the beam search algorithm for RNN-T models is given as O(B * T * U * C), where B is the beam size, T is the number of audio frames in the input, U is the max length of the output, and C is the cost of the forward pass. PRISM method adds a lookup cost to memory in each step of the beam expansion. If the cost of lookup is P, then the overall complexity would become O(B * T * U * (C + P)).
> We use the state-of-the-art nearest neighbor index from Johnson et al [1], which can do a fast look-up to minimize the latency.
>
> [1] Johnson, Jeff, Matthijs Douze, and Hervé Jégou. "Billion-scale similarity search with GPUs." IEEE Transactions on Big Data 7.3 (2019): 535-547.
>
>
> Q: `In Table 2, are the results of RNN-T+TCPGen and RNN-T+Deep Cont. taken from literature, or regenerated?`
>
> A: Yes, these numbers are cited from the respective papers. The models from these papers are not open source, and Deep Contextual RNN-T is trained on proprietary data internal to the organization which makes reproducibility challenging.

---

### Meta-Review · Area_Chair_vUXb · 2023-09-15

**Recommendation:** 5

**Metareview:**

This paper presents a novel method that allows adapting ASR systems (transducers and enc-dec) to dictionaries containing rare words and phrases without retraining the original ASR model. The method uses kNN retrieval and TTS to obtain speech for rare words. The idea is well-presented and achieves consistent gains on three datasets. The authors will also release their entity-rich speech dataset to facilitate further research into dictionary-based ASR adaptation.

The authors addressed all reviewers' comments, especially regarding computation cost and comparison with additional methods. All reviewers are excited about this paper and assigned high soundness scores.

---

### Decision · Program_Chairs · 2023-10-07

**Decision:**

Accept-Main

**Comment:**

This paper presents a novel method that allows adapting ASR systems (transducers and enc-dec) to dictionaries containing rare words and phrases without retraining the original ASR model. The method uses kNN retrieval and TTS to obtain speech for rare words. The idea is well-presented and achieves consistent gains on three datasets. The authors will also release their entity-rich speech dataset to facilitate further research into dictionary-based ASR adaptation.

The authors addressed all reviewers' comments, especially regarding computation cost and comparison with additional methods. All reviewers are excited about this paper and assigned high soundness scores.